# Assessment of Virtual Reality as a Didactic Resource in Higher Education

Diego Vergara [1,*], Álvaro Antón-Sancho [2], Jamil Extremera [3] and Pablo Fernández-Arias [1]

1 Department of Mechanical Engineering, Catholic University of Ávila, C/Canteros, s/n, 05005 Ávila, Spain; pablo.fernandezarias@ucavila.es
2 Department of Mathematics and Experimental Science, Catholic University of Ávila, C/Canteros, s/n, 05005 Ávila, Spain; alvaro.anton@ucavila.es
3 Department of Computer Science and Automatics, University of Salamanca, Plaza de los Caídos, s/n, 37008 Salamanca, Spain; jamil.extremera@usal.es
* Correspondence: diego.vergara@ucavila.es

**Abstract:** Given that the university teachers with more experience in the use of virtual reality are those corresponding to the areas of Health Sciences and of Engineering and Architecture, this article analyzes the assessment these teachers make about virtual reality as a teaching resource in their respective disciplines. The study uses a questionnaire that assesses the technical aspects and future projection of virtual reality, its drawbacks and the perception of the different dimensions of the participants' knowledge about virtual reality and its didactic employability. The questionnaire was answered by a sample of 423 university teachers of different genders, ages, academic level and teaching experience, whose teaching activity is developed in various Latin American universities in the area of Health Sciences or in the area of Engineering/Architecture. Their answers have been analyzed descriptively and Spearman's r statistics and the Multifactor ANOVA test have been used to verify the existence of significant differences in their evaluations for the different variables considered, cross-referencing them with the field of knowledge. Within the main results, gaps by area, years of teaching experience and academic level in the participants' evaluations have been identified and discussed.

**Keywords:** virtual reality; didactic resource; higher education

## 1. Introduction

Immersive virtual reality (IVR) can be defined as "the sum of the hardware and software systems that seek to perfect an all-inclusive, sensory illusion of being present in another environment" [1], with two central elements being interactivity and the sensation of presence in the virtual environment [2]. This sensation of presence in the virtual environment is achieved mainly through the use of head-mounted displays (HMDs) that place screens projecting stereoscopic images of three-dimensional environments in front of the user's eyes [3], while interactivity (i.e., real-time manipulation of the virtual environment by the user [4]) is achieved mainly through the use of gadgets that usually accompany the most widely used commercial virtual reality (VR) kits [5,6]. In addition, there are other control devices, e.g., haptic gloves, Leap Motion system, joysticks, physical objects simulating tools, etc. [7]. VR is a technology that appeared in the second half of the 20th century, with the work of Sutherlan [8] being considered the starting point of this technology [9]. Interest in it has varied over time, having experienced a great increase in the last decade if we consider the growing number of academic papers published annually that contain the expression "virtual reality" [7,10].

On the other hand, augmented reality (AR) makes it possible to combine virtual objects and real objects [11], so that virtual objects are spatially located in a real environment [12,13]. Until about a decade ago, AR was linked to the use of cameras connected to a computer

(see, for example, the work of Núñez et al. [14]). It is now mainly linked to the use of mobile devices (smartphones and tablets) and HMDs (i.e., HoloLens [15]) [16]. Unlike VR, which replaces the real world with a virtual world, AR "augments" the real world without replacing it [17]. An example of a current application based on the use of AR is the popular game Pokémon GO [18], in which the player must capture virtual characters found in certain locations in the real environment [19].

Mixed reality (MR) was described by Milgram and Kishino [20] as a continuum of virtual experiences ranging from the fully virtual world to the real world, passing through various degrees of reality and virtuality. MR can be considered as the blending of real and virtual environments, employing VR and AR technologies [21]. This type of technology is mainly based on the use of HMDs such as HoloLens, which allow the user to interact both with virtual elements superimposed on the real environment and to be immersed in fully virtual environments [15].

The evolution of information and communications technologies (ICT) has brought with it the expansion of the use of technologies such as VR, AR or MR within the educational sector, and there are currently many scientific works in which some of these technologies are used to improve the teaching-learning process of various disciplines [7]. Thus, some examples of the use of this type of technology include, among many others, the virtual chemistry laboratory described in the work of Su and Cheng [22], the Mandarin language learning platform by Le-gault et al. [23], the virtual anatomy atlas for surgical training by Weyhe et al. [24], the dental morphology learning application by Juan et al. [25], the history learning application by Utami et al. [26] or the virtual laboratories for learning materials science and engineering by Vergara et al. [27]. In the studies included in this type of work, it is common to evaluate parameters such as the impact of these educational tools on learning (as occurs, for example, in the work of Tarng et al. [28]) or the opinion that students have about them (as occurs, for example, in the work of Vergara et al. [29]), i.e., the studies focus on the student. Since the teaching-learning process necessarily consists of a teaching party (teachers) and a learning party (students), teacher-centered studies (such as that of Tzima et al. [30] or Cuccurullo et al. [31]) become necessary to obtain data to build as broad a view as possible about the use of VR, AR and MR in education.

A search in databases shows that there is a growing trend in the use of technologies such as VR since the beginning of the 21st century [3]. In fact, a search in the Scopus database containing the terms "virtual reality", "augmented reality" or "mixed reality" in the title, abstract or keywords shows that, since 2001, the number of papers published has increased with respect to the previous year in almost every year of the period, with a more pronounced growth being observed in the second half of the decade between 2010 and 2015 (Figure 1). This same trend can be observed when the results are restricted to the subject areas of Engineering and Health Sciences, although the increase in Health Sciences since 2015 is less pronounced (Figure 1). Therefore, it is possible to state that for several years there has been a growing interest in the use of technologies based on VR, AR and MR in the knowledge areas related to Engineering and Health Sciences. Note that, in 2020, both the total number of papers and the proportion of them belonging to the field of Engineering decrease in relation to 2019. This fact, which should be studied in subsequent years, may be linked to the COVID-19 pandemic situation, which has slowed down the use of resources that require the presence of students and teachers in educational centers (as occurs with the IVR since, to be used, the student must employ a series of headsets and gadgets that are usually found in a classroom at their university or institution).

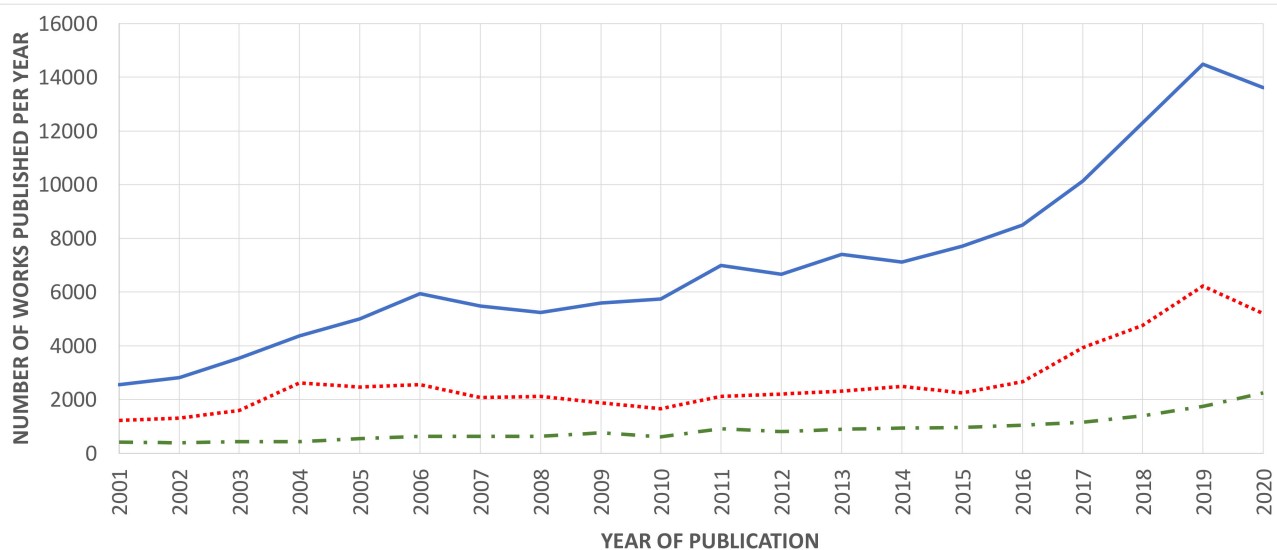

**Figure 1.** Number of papers published each year containing the terms "virtual reality", "augmented reality" or "mixed reality" in their title, abstract or keywords, indexed in Scopus from 2001 to 2020. The total number of papers (all subject areas) is shown, as well as the number of papers belonging to "Medicine" and "Engineering", respectively.

Thus, the use of educational tools based on the use of VR, AR and MR in education is a consolidated and expanding phenomenon, and there are indications that these technologies are of growing interest in the areas of Engineering and Health Sciences, as these are fields of knowledge whose concepts are more naturally representable through these technologies. Indeed, in Health Sciences, it is used for the representation of organs and the learning of their structures, the acquisition of manual medical skills (e.g., in surgery) and the use of certain equipment [32]. In technical education, this type of technology is often used for the design of laboratory and equipment simulator environments (mechanical or electrical, for example) [33,34]. Therefore, this paper aims to obtain information regarding the perception and knowledge that Engineering and Health Sciences teachers have about educational tools based on VR (both IVR and non-immersive VR -NIVR-), AR and MR using a study focused on a sample of university teachers belonging to these areas of knowledge.

This study is focused on the geographical area of Latin America, which is experiencing a strong growth of investment in digital technologies, and mainly in VR, because it is the strongest digital market in this area at present (the Compound Annual Growth Rate -CAGR- of VR in this area is 49.61% in the period from 2020 to 2025, according to a market research report by the company MarketsandMarkets [35]). Although this study is especially concerned with VR technology, it also analyzes the opinion of which technology (VR, AR and MR) university faculties consider to have the greatest future projection in the education sector. For the two areas of knowledge studied, the work differentiates by gender, age, teaching experience and academic level. In this way, it will be possible to identify whether there are gaps based on gender, age or level in the use of virtual reality in the classroom, which will make it possible to suggest the need for universities to propose specific training measures in this regard, and highlight which population should be targeted. Likewise, the distinction between Health Sciences and Engineering will make it possible to understand to what extent the latter's knowledge of technology influences the didactic employability of virtual reality.

## 2. Materials and Methods

The following study is descriptive and consists of the analysis of the answers given by a sample of university teachers of Health Sciences and Engineering/Architecture to a questionnaire on the didactic use of VR in the classroom. The questionnaire takes into account a set of independent or descriptive variables and a family of dependent or evaluative variables (Figure 2). The descriptive variables serve to delimit the sociological and academic profile of the participants. The participants were selected by means of a non-probabilistic convenience sampling procedure.

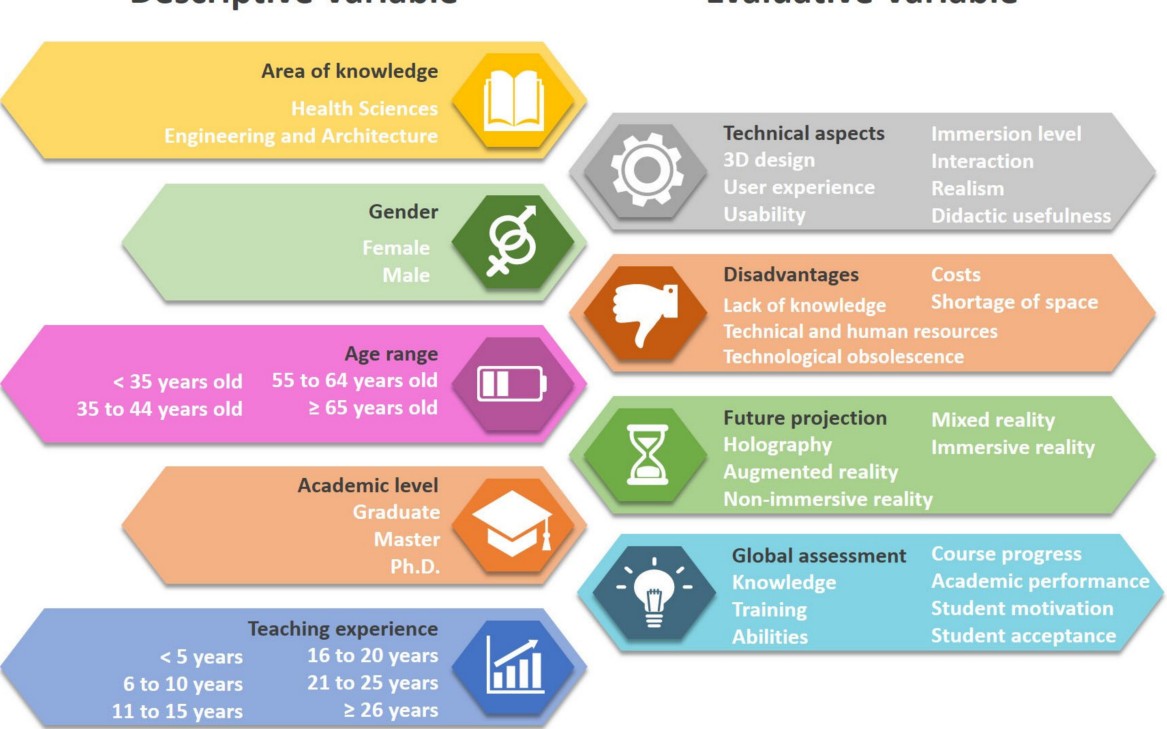

**Figure 2.** Descriptive and evaluative variables.

A total of 423 teachers from 17 Latin American countries (Argentina, Bolivia, Brazil, Chile, Colombia, Ecuador, El Salvador, Guatemala, Honduras, Mexico, Nicaragua, Panama, Paraguay, Peru, Dominican Republic, Uruguay and Venezuela) participated in the study and answered the questionnaire freely and anonymously. These participants constitute the totality of the target population that was contacted. In light of World Bank data on social inequality, expressed in the Gini Index (0 expresses no inequality and 100 extreme inequality), the countries considered suffer from high levels of inequality (for example, Panama has 49.8, Peru 45.1 and Argentina 42.9). Assuming that university teachers are part of the most favored social sector, the results obtained in this work cannot be extrapolated to the bulk of the societies within the respective countries, but rather to university teachers in the areas of Health Sciences and Engineering in other more developed countries, whether European or North American. The distribution of the participants according to the values of the different independent variables is shown in Table 1.

The most represented area of knowledge is Engineering, which accounts for slightly less than two thirds of the sample. The goodness-of-fit test confirms that the distribution is not homogeneous when differentiated by area of knowledge (Chi-square = 10.7870, $p$-value = 0.0010). The distribution of participants by gender is approximately homogeneous, as confirmed by the goodness-of-fit test (Chi-square = 0.0638, $p$-value = 0.8005). This is not the case when differentiating by age (Chi-square = 47.05, $p$-value = 0.0000). The central age bands concentrate the highest proportions of participants, especially the 45 to

54 years of age range, while the extreme age ranges (the youngest and the longest-lived) are in the minority. As for the academic level, the highest proportion is accumulated in the Master's degree category, which congregates more than half of the participants, followed by the PhDs. Nor for this variable is the distribution of the sample homogeneous (Chi-square = 65.489, *p*-value = 0.0000). The distribution of participants by years of teaching experience is more balanced and can be assumed to be homogeneous (Chi-square = 2.8723, *p*-value = 0.7197).

**Table 1.** Descriptive variables.

| Descriptive Variable | Character | Values | Proportion in the Sample |
|---|---|---|---|
| Area of knowledge | Dichotomous | Health Sciences | 36% |
| | | Engineering and Architecture | 64% |
| Gender | Dichotomous | Female | 49% |
| | | Male | 51% |
| Age range | Polytomous | <35 years old | 11% |
| | | 35 to 44 years old | 22% |
| | | 45 to 54 years old | 40% |
| | | 55 to 64 years old | 21% |
| | | ≥65 years old | 6% |
| Academic level | Trichotomous | Graduate | 14% |
| | | Master | 65% |
| | | Ph.D. | 21% |
| Teaching experience | Polytomous | ≤5 years | 15% |
| | | 6 to 10 years | 19% |
| | | 11 to 15 years | 16% |
| | | 16 to 20 years | 15% |
| | | 21 to 25 years | 14% |
| | | >26 years | 21% |

The dependent variables measure, on a Likert scale from 1 to 5, is the participants' perception of 24 different aspects of VR and its didactic use at the university. These variables make up the 24 items of the questionnaire. In the answers, 1 is identified with the worst evaluation and 5 with the best. The 24 items in the questionnaire are grouped into four scales which correspond to four different dimensions of VR: technical aspects, identification and assessment of its disadvantages, future projection as a teaching resource and an overall assessment scale of the knowledge, training and skills of teachers in the use of VR and its influence on their classroom activity and the academic performance it induces. The characteristics of the dependent variables are detailed in Table 2. The internal consistency of the questionnaire has been validated by means of Cronbach's alpha parameter, which has been computed for each scale and is also shown in Table 2. All the parameters are above 0.7, indicating that all the scales enjoy adequate levels of consistency.

The analysis of the results combines a descriptive part and an inferential and correlation analysis part. First, the descriptive statistics of all the dependent variables and the overall statistics of the different scales of the questionnaire are analyzed. Next, Spearman's r coefficient is used to identify which of the independent variables turn out to be discriminant of which dependent variables, in the sense that, when differentiating by that descriptive variable, the monotonicities of the corresponding dependent variables are explained by the

descriptive variable. Likewise, the results of the dependent variables are analyzed when differentiating by the discriminant variables identified above. Finally, the statistics of the results of the questionnaire are presented for the different scales differentiated by each of the independent variables crossed with the knowledge area variable and the Multifactor ANOVA (hereafter, MANOVA) tests are applied to check if there are significant differences between the mean answers. Bartlett's test is also used to identify significant differences between standard deviations. All tests are performed with a statistical significance of 0.05.

**Table 2.** Evaluative variables.

| Evaluative Variable | Cronbach's Alpha | Values |
|---|---|---|
| Technical aspects of VR | 0.8566 | 3D design<br>User experience<br>Usability<br>Immersion level<br>Interaction<br>Realism<br>Didactic usefulness |
| Disadvantages of VR | 0.7427 | Costs<br>Shortage of space<br>Technical and human resources<br>Lack of knowledge<br>Technological obsolescence |
| Future projection | 0.8286 | Holography<br>Augmented reality<br>Mixed reality<br>Immersive reality<br>Non-immersive reality |
| Global assessment of VR | 0.7016 | Knowledge<br>Training<br>Abilities<br>Academic performance<br>Course progress<br>Student motivation<br>Student acceptance |

## 3. Results

### 3.1. Global Results

The mean and standard deviation statistics for each scale and for each of its items are shown in Table 3. The participants give high or very high mean ratings to the VR in terms of its technical aspects, especially with regards to its didactic usefulness. The scale of technical aspects is, in fact, the one with the highest overall mean value. It is also the one with the lowest deviation, which shows that the answers of the participants are more unanimously agreed upon than on the other scales. In future projection, the evaluations are intermediate-high, with scores between 3 and 4 for all the items studied. The techniques whose future projection are the worst rated are holography and NIVR. However, these two technologies are the ones with the highest deviation, which shows greater heterogeneity and, therefore, a more poorly formed concept compared to the rest of the techniques.

Despite the above, the participants perceive an intermediate-high degree of disadvantages in VR as a didactic resource, at a medium level. In this sense, there are no notable differences between the different disadvantages proposed in the questionnaire and the deviations of the results being high (around a third of the mean value), but similar in all of them.

With regards to the overall assessment of VR as a teaching resource, the participants have an intermediate-low self-concept of their knowledge and the training they have

received in this respect, although they consider that they have more skill than knowledge (about six tenths more, on average). However, the evaluations increase above a score of 4, on average, when the didactic effectiveness of VR is judged, both in the progress of the classes and in the motivation, acceptance and academic results of the students. The deviations of the answers are clearly smaller for the latter questions, which indicates a greater agreement among the participants in this respect and, consequently, a more solidly formed concept. All this means that the overall mean value for this scale, between 3 and 4, is given with high standard deviation, due to the differences between the answers to the first three items compared to the last five.

**Table 3.** Means and standard deviations of the different items of the questionnaire and of each of the scales.

| Scale | Value Variable | Mean | St.D. | Global Mean | Global St.D. |
|-------|----------------|------|-------|-------------|--------------|
| Technical aspects | 3D design | 4.03 | 1.01 | 4.19 | 0.91 |
| | User experience | 4.16 | 0.92 | | |
| | Usability | 4.17 | 0.87 | | |
| | Immersion level | 4.00 | 0.91 | | |
| | Interaction | 4.28 | 0.90 | | |
| | Realism | 4.16 | 0.89 | | |
| | Didactic usefulness | 4.54 | 0.78 | | |
| Disadvantages | Costs | 3.83 | 1.18 | 3.57 | 1.27 |
| | Shortage of space | 3.74 | 1.29 | | |
| | Tecnical and human resources | 3.76 | 1.13 | | |
| | Lack of knowledge | 3.79 | 1.19 | | |
| | Technological obsolescence | 3.71 | 1.20 | | |
| Future projection | Augmented reality | 4.10 | 0.89 | 3.94 | 1.01 |
| | Mixed reality | 4.09 | 0.92 | | |
| | Immersive reality | 4.10 | 0.97 | | |
| | Non-immersive reality | 3.67 | 1.02 | | |
| Global assessment | Knowledge | 2.70 | 1.09 | 3.56 | 1.28 |
| | Training | 2.28 | 1.29 | | |
| | Habilities | 3.29 | 1.13 | | |
| | Academic performance | 4.26 | 0.88 | | |
| | Course progress | 4.25 | 0.84 | | |
| | Student motivation | 4.39 | 0.73 | | |
| | Student acceptance | 4.13 | 0.97 | | |

Student's *t*-test of comparison of the means of the four scales with three degrees of freedom ($t = 75.768$, *p*-value = 0.0000) confirms that there are statistically significant differences between the mean values of the different scales of the questionnaire. Likewise, the Levene's test statistics for comparison of standard deviations (F = 85.824, *p*-value = 0.0000) indicate that there is also a significant distinction between the dispersions of the different scales, such that the answers are more heterogeneous on the VR drawbacks and overall assessment of VR scales than on the technical aspects and future projection scales.

An analysis of the influence of the independent variables on the behavior of the dependent variables was carried out. Since the dependent variables are ordinal, Spearman's r correlation coefficient was chosen for this purpose. The values of this statistic range from $-1$ to $+1$ and indicate whether the corresponding descriptive variable is significantly discriminant of the answers to each item in the questionnaire. Values of the coefficient

close to +1 indicate positive monotonicity, while values close to −1 indicate negative monotonicity and 0 indicates no monotonicity. Table 4 shows the values of Spearman's r coefficients when all the dependent variables of the questionnaire are cross-referenced with all the independent variables. In addition, we have highlighted those coefficients that yield a paired *p*-value lower than the 0.05 significance level, which allows us to assume that the value of the coefficient is statistically significant and not due to chance.

**Table 4.** Spearman's r coefficients for all the assessment variables crossed by all the descriptive variables.

| Variable | Area of Knowledge | Gender | Age | Academic Level | Experience |
|---|---|---|---|---|---|
| 3D design | −0.07 | −0.27 * | 0.10 | −0.08 | 0.11 |
| User experience | 0.27 * | 0.02 | −0.76 * | −0.23 | 0.21 |
| Usability | 0.26 | 0.11 | −0.40 | 0.26 | −0.06 |
| Immersion level | −0.40 * | −0.01 | −0.39 | −0.12 | 0.25 |
| Interaction | −0.02 | −0.12 | 0.04 | −0.09 | 0.18 |
| Realism | −0.26 | −0.15 | 0.12 | 0.03 | −0.26 |
| Didactic usefulness | −0.04 | −0.01 | −0.11 | −0.17 | −0.43 * |
| Costs | −0.39 * | 0.21 | −0.33 | 0.18 | −0.09 |
| Shortage of space | 0.02 | 0.02 | −0.41 | 0.23 | −0.47 * |
| Tecnical and human resources | −0.15 | 0.06 | 0.59 | −0.01 | 0.13 |
| Lack of knowledge | −0.08 | 0.05 | 0.06 | −0.25 | 0.40 |
| Technological obsolescence | −0.24 | 0.04 | −0.38 | 0.05 | 0.23 |
| Holography | 0.24 | −0.14 | −0.40 | 0.16 | −0.08 |
| Augmented reality | 0.06 | −0.08 | 0.37 | −0.09 | −0.35 |
| Mixed reality | 0.32 * | 0.02 | 0.03 | −0.04 | −0.36 |
| Immersive reality | 0.16 | 0.03 | 0.47 | 0.05 | 0.11 |
| Non-immersive reality | 0.10 | 0.05 | 0.25 | 0.02 | 0.35 |
| Knowledge | −0.06 | −0.30 * | −0.21 | −0.08 | −0.08 |
| Training | 0.09 | −0.02 | −0.01 | −0.19 | 0.26 |
| Habilities | −0.11 | 0.17 | 0.23 | −0.04 | 0.65 * |
| Academic performance | 0.05 | 0.02 | −0.36 | −0.13 | 0.10 |
| Course progress | 0.11 | 0.07 | 0.43 | −0.04 | 0.16 |
| Student motivation | −0.01 | −0.09 | −0.06 | −0.11 | −0.11 |
| Student acceptance | −0.17 | 0.01 | −0.10 | −0.39 * | −0.16 |

* $p < 0.05$.

The values of the coefficients in Table 4 are small in absolute value, which indicates that, in general, there are few crossings in which the value variable is explained by the corresponding descriptive variable. When a *p*-value of less than 0.05 is also required, the number of such crossovers is further reduced, although it is clear that the knowledge area variable is the most discriminative. Next, the influence of the independent variables on the discrimination of the dependent variables is discussed, both in terms of the monotonicity of the distributions and the behavior of the means and standard deviations for the different scales.

### 3.2. Area of Knowledge

It was observed in Table 4 that the descriptive variable of the area of knowledge is the most discriminative of all for the technical aspects scale and that it is also discriminative for some items of other scales. Figure 3 shows the frequencies of the answers to all these items when differentiated by area of knowledge. Regarding technical aspects (user experience and immersion), it can be seen that the lowest answers (1 to 3) are more frequent in Health Sciences, while responses of 4 and 5 are more frequent in Engineering. In this observation, the frequency of the answer 4 in the item on user experience should be excepted, the proportion of which is notably higher in Health Sciences. At the mean level, the overall ratings in Health Sciences (4.04 for user experience and 3.82 for degree of immersion) are slightly lower than the respective means in Engineering (4.32 for user experience and 4.10 for degree of immersion). However, the *t*-test statistics show that the differences between the means are not statistically significant for either user experience ($t = -1.193$, *p*-value = 0.2356) or degree of immersion ($t = -1.6548$, *p*-value = 0.1015).

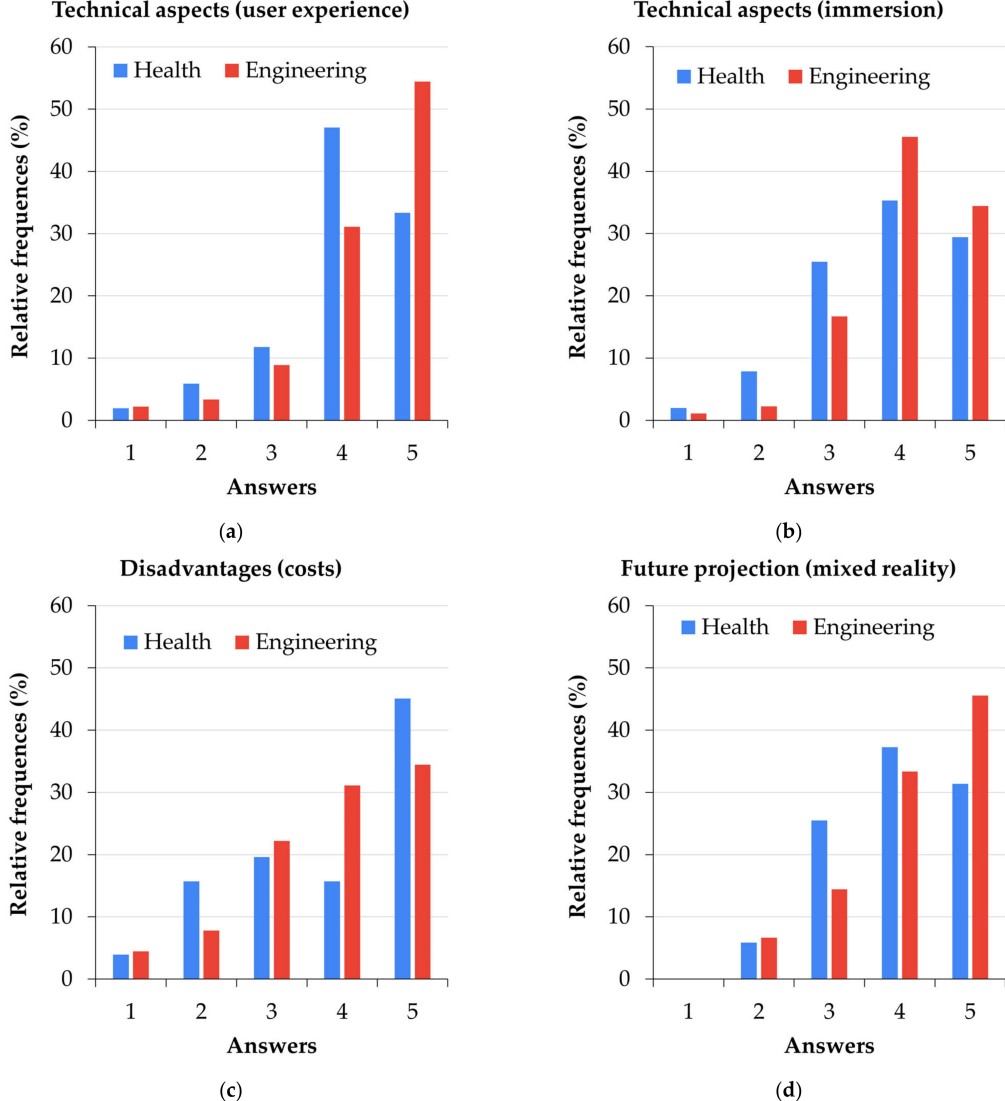

**Figure 3.** Answers on (**a**) user experience and (**b**) immersion, (**c**) costs and (**d**) future projection of MR, differentiated by area of knowledge.

For the item that values costs, within the inconvenience scale, Engineering participants accumulate proportions of responses that increase with the value, from 1 to 5, of the answer.

In the case of Health Sciences, answer 2 is relatively frequent (especially in comparison with answer 1), answer 4 is less frequent than answer 3 and the same as answer 2, and it is answer 5 that accumulates the highest relative frequency. This translates into very similar mean values for both areas in this item (3.82 for Health Sciences and 3.83 for Engineering), and the *t*-test does not allow us to assume significant differences between the two mean values ($t = -0.0457$, *p*-value = 0.9636).

With respect to the item assessing MR, neither of the two areas provided answers of value 1; the area of Engineering exceeded that of Health Sciences in the frequency of value 5 in a very notable way and of value 2 in a more discrete way. This means that the mean values of the participants in Engineering (4.18) slightly exceed the mean in Health Sciences (3.94), although the difference is not statistically significant either, in light of the *t*-test statistics ($t = -1.4848$, *p*-value = 0.1406).

The mean values of the participants, differentiated by areas, are shown in Table 5. Engineering teachers outperform Health Sciences teachers on all scales except for the VR disadvantages scale. In any case, the differences are very small in all the scales. The largest of these differences is on the technical aspects scale (16 hundredths of a point). In it, the *t*-test shows that there is a statistically significant gap between the two areas of knowledge studied. Furthermore, the dispersion in this scale is lower in Engineering than in Health Sciences (Table 6), although it is not a statistically significant difference. For the rest of the scales, it is not possible to speak of a significant gap by area of knowledge. However, in the VR disadvantages scale, the *p*-value of the Levene's test indicated in Table 6 shows that, although in Health Sciences there is a greater perception of these disadvantages, there is also less unanimity, because the heterogeneity of the answers is greater than in the area of Engineering.

**Table 5.** Mean answers and *t*-test statistics when differentiated by area of knowledge.

| Scale | Health Sciences | Engineering and Architecture | Student *t* | *p*-Value |
|---|---|---|---|---|
| Technical | 4.09 | 4.25 | 7.2409 | 0.0073 * |
| Future | 3.92 | 3.95 | 0.1250 | 0.7238 |
| Disadvantages | 3.58 | 3.56 | 0.0407 | 0.8403 |
| Assessment | 3.56 | 3.57 | 0.0099 | 0.9208 |

* $p < 0.05$.

**Table 6.** Standard deviations and Levene's test statistics when differentiated by area of knowledge.

| Scale | Health Sciences | Engineering and Architecture | Levene F | *p*-Value |
|---|---|---|---|---|
| Technical | 0.95 | 0.89 | 0.6926 | 0.4055 |
| Future | 0.95 | 1.05 | 2.9391 | 0.0869 |
| Disadvantages | 1.32 | 1.24 | 5.5700 | 0.0185 * |
| Assessment | 1.26 | 1.30 | 0.0555 | 0.8138 |

* $p < 0.05$.

### 3.3. Gender

It has been shown that gender has a certain discriminating power in the evaluation of 3D design, within the scale of technical aspects. Figure 3 shows that the tendency is for females to have a higher proportion of maximum scores than males, while males offer higher proportions of intermediate and low ratings (from 1 to 3). The mean answers for the variable considered (4.14 for females and 3.93 for males) reinforce the above idea, although the *t*-test statistics ($t = 1.2535$, *p*-value = 0.2121) do not find significant differences between

them. A gender discriminatory character has also been observed in the evaluation of the students' acceptance of VR. In this regard, females accumulate the highest proportion of rating 5, and males do so with the rest of the ratings (Figure 4). At the mean level, there is a slight superiority of the answer of females (4.16) with respect to males (4.10), but the *t*-test does not support this difference as significant ($t = 0.3779$, *p*-value = 0.7061).

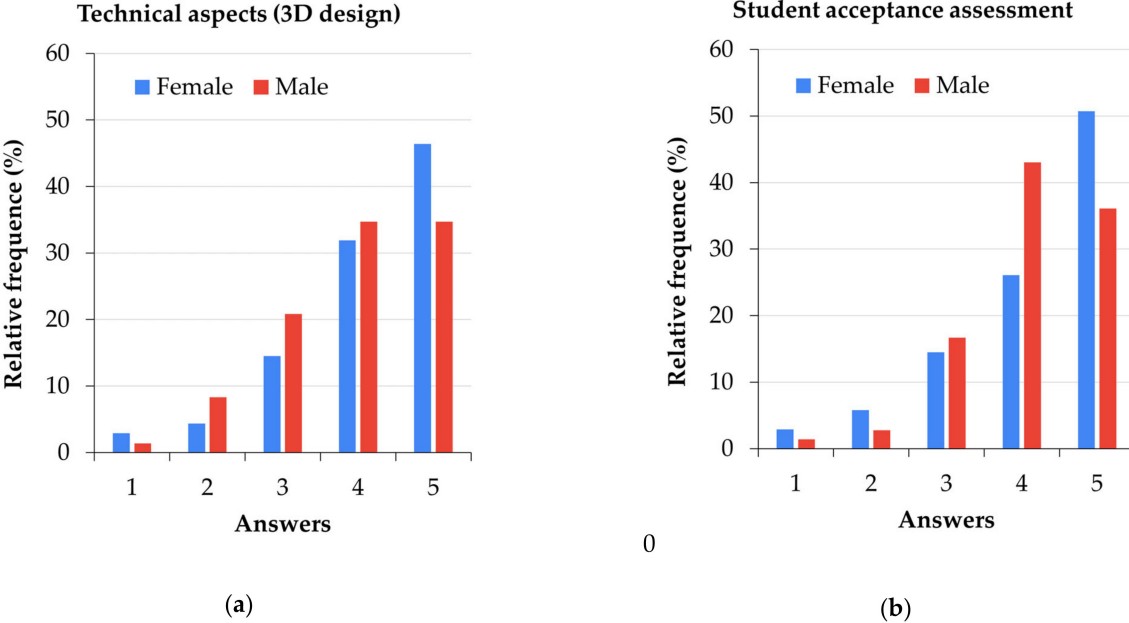

(**a**)        (**b**)

**Figure 4.** Answers on (**a**) technical aspects (3D design) and (**b**) student acceptance of VR differentiated by gender.

Table 7 shows the mean answers by areas of knowledge of the scales, differentiated by gender. Females report a better mean score than males except in the identification of disadvantages. Females score better on technical aspects in both fields of knowledge. Among specialists in Engineering, males identify more drawbacks and perceive greater future projection, while the scores show minimal differences between genders among specialists in Health Sciences. In any case, the ANOVA test statistics do not find a gender gap in any area. Furthermore, homoscedasticity is found in all the scales (Table 8) except for technical aspects and global assessment. Regarding the latter scale, females show greater heterogeneity of answers than males in both areas. In technical aspects, males give more homogeneous answers in Health Sciences, and females do so in Engineering.

**Table 7.** Mean answers and MANOVA test statistics when differentiated by gender.

| Scale | Female | | Male | | MANOVA | *p*-Value |
|---|---|---|---|---|---|---|
| | Health | Engineering | Health | Engineering | | |
| Technical | 4.12 | 4.38 | 4.03 | 4.17 | 0.8788 | 0.3488 |
| Future | 3.92 | 3.81 | 3.92 | 4.04 | 1.9345 | 0.1647 |
| Disadvantages | 3.59 | 3.50 | 3.57 | 3.60 | 0.3455 | 0.5569 |
| Assessment | 3.52 | 3.64 | 3.63 | 3.52 | 1.8429 | 0.1749 |

**Table 8.** Standard deviations and Bartlett test statistics when differentiated by gender.

| Scale | Female | | Male | | K-squared | *p*-Value |
|---|---|---|---|---|---|---|
| | Health | Engineering | Health | Engineering | | |
| Technical | 0.97 | 0.80 | 0.91 | 0.93 | 10.847 | 0.0126 * |
| Future | 0.99 | 1.12 | 0.88 | 0.99 | 7.4829 | 0.0580 |
| Disadvantages | 1.31 | 1.25 | 1.34 | 1.23 | 1.5827 | 0.6633 |
| Assessment | 1.32 | 1.40 | 1.15 | 1.22 | 10.541 | 0.0145 * |

* $p < 0.05$.

### 3.4. Age Range

Table 4 shows that the only rating variable for which age is discriminative is user experience within the technical aspects. The frequencies of the answers for this variable, differentiated by age, are shown in Figure 5. It can be seen that the youngest participants accumulate the highest proportions of the highest ratings (4 and 5), while the oldest participants, although they give the highest proportion in value 4, give the lowest in value 5. Although this is a notable difference in absolute terms, the ANOVA test statistics do not allow us to assume that it is statistically significant ($F = 2.463$, $p$-value = 0.0609).

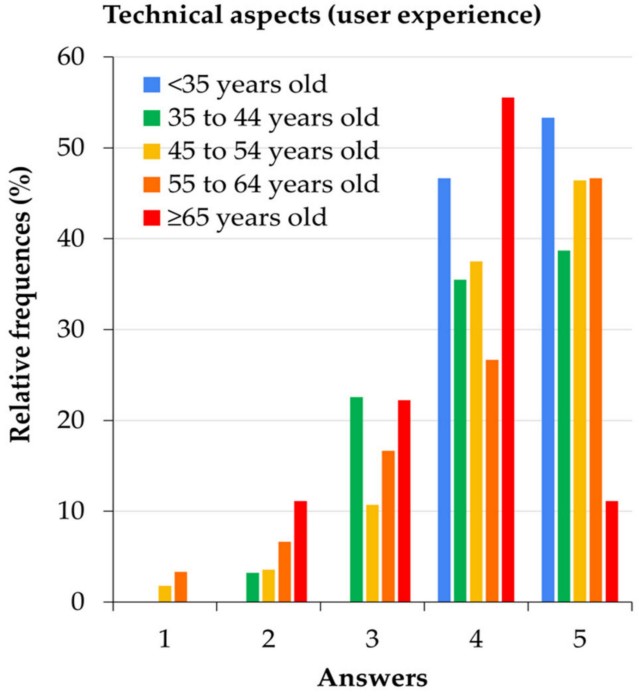

**Figure 5.** Answers on technical aspects (user experience) of VR differentiated by age range.

The mean answers for the different scales by areas of knowledge when differentiated by age are shown in Table 9. The MANOVA test statistics show that it cannot be assumed, also for the age variable, that there are significant gaps in any scale. Regarding the deviations shown in Table 10, Bartlett's test shows that there are differences by age and area in the scales of technical aspects and global assessment. Specifically, among those over 65 years of age, the distribution of the answers of the Health Sciences teachers on these scales is notably more homogeneous than that of the Engineering teachers, which shows a more diffuse or a less certain concept on the part of the latter.

**Table 9.** Mean answers and MANOVA test statistics when differentiated by age range.

| Scale | <35 | | 35 to 44 | | 45 to 54 | | 55 to 64 | | ≥65 | | MANOVA | *p*-Value |
|---|---|---|---|---|---|---|---|---|---|---|---|---|
| | H. | E. | H. | E. | H. | E. | H. | E. | H. | E. | | |
| Technical | 4.04 | 4.45 | 3.77 | 4.09 | 4.15 | 4.30 | 4.21 | 4.36 | 3.86 | 3.98 | 0.6162 | 0.6511 |
| Future | 3.45 | 4.11 | 3.98 | 3.88 | 3.91 | 3.96 | 4.03 | 4.15 | 4.00 | 3.46 | 2.2121 | 0.0662 |
| Disadvantages | 3.20 | 3.51 | 3.43 | 3.63 | 3.58 | 3.59 | 3.79 | 3.63 | 3.40 | 3.11 | 0.6630 | 0.6178 |
| Assessment | 3.45 | 3.66 | 3.59 | 3.53 | 3.57 | 3.65 | 3.51 | 3.51 | 3.94 | 3.30 | 1.0361 | 0.3373 |

**Table 10.** Standard deviations and Bartlett test statistics when differentiated by age range.

| Scale | <35 | | 35 to 44 | | 45 to 54 | | 55 to 64 | | ≥65 | | K-Squared | *p*-Value |
|---|---|---|---|---|---|---|---|---|---|---|---|---|
| | H. | E. | H. | E. | H. | E. | H. | E. | H. | E. | | |
| Technical | 0.88 | 0.77 | 0.81 | 0.96 | 0.96 | 0.88 | 1.03 | 0.77 | 0.53 | 0.95 | 21.536 | 0.0105 * |
| Future | 0.83 | 1.01 | 0.89 | 1.06 | 1.00 | 1.10 | 0.96 | 0.97 | 0.47 | 0.85 | 14.327 | 0.1111 |
| Disadvantages | 1.06 | 1.20 | 1.28 | 1.22 | 1.45 | 1.24 | 1.27 | 1.16 | 0.52 | 1.43 | 16.760 | 0.0526 |
| Assessment | 0.99 | 1.28 | 1.32 | 1.36 | 1.34 | 1.34 | 1.25 | 1.17 | 0.25 | 1.16 | 43.793 | 0.0000 * |

* $p < 0.05$.

There is a certain association between gender and area of knowledge, in the sense that the most frequent gender in Health Sciences is female and in Engineering it is male. However, this association is weak, although statistically significant, as indicated by the value of Cramer's V parameter, which values this dependence for nominal variables in a range between 0 (no dependence) and 1 (strong dependence). In this case, the parameter reaches a value of V = 0.2375, with a *p*-value = 0.0046. For this reason, there is a certain divergence of results in the analysis of both variables.

*3.5. Academic Level*

It was found that the academic level of the participants is statistically discriminative of the motivation induced in the students, within the scale of global evaluations of VR as a didactic resource. In this regard, Figure 6 shows the frequencies of the answers to this item when differentiated by academic level. The highest scores (4–5) are concentrated to a greater extent among PhDs and, to a lesser extent, among Masters. Among graduates, although the mode is 5, there is a very high proportion of responses of 3. This means that the highest average answer is provided by PhDs (4.55), followed by Masters (4.38) and, lastly, by graduates (4.20). However, the differences between these mean values cannot be assumed to be statistically significant in light of the ANOVA test statistics (F = 1.4725, *p*-value = 0.2406).

The MANOVA test (Table 11) shows that there are significant differences between the mean answers of graduates, Masters and PhDs for the scales of technical aspects and disadvantages in the two areas of knowledge studied. Among Engineering teachers, PhDs give the best evaluation of the technical aspects of VR and the lowest of its disadvantages. In Health Sciences, on the other hand, it is the graduates who rate the technical aspects most positively, although they are also the ones who identify the disadvantages most positively. As for the deviations (Table 12), homoscedasticity can be assumed for all the scales.

The academic level of the participants and their age range do not have, for the participants surveyed, a relationship of mutual dependence, as indicated by Cramer's V parameter (V = 0.1164, *p*-value = 0.1693), which indicates that the association between the two variables is not statistically significant and, in any case, would be very weak. Therefore, although it might be expected that the PhDs would be the oldest participants and, consequently, that the results for the descriptive variable of academic level would

be similar to those discussed for the age variable. However, this does not occur because academic level does not seem to be influenced by age.

### Assessment of student motivation

**Figure 6.** Answers on student motivation (global assessment) of VR differentiated by academic level.

**Table 11.** Mean answers and MANOVA test statistics when differentiated by academic level.

| Scale | Graduate | | Master | | Ph.D. | | MANOVA | *p*-Value |
|---|---|---|---|---|---|---|---|---|
| | Health | Engineering | Health | Engineering | Health | Engineering | | |
| Technical | 4.40 | 4.04 | 4.01 | 4.26 | 4.04 | 4.24 | 6.9020 | 0.0011 * |
| Future | 3.91 | 3.84 | 3.98 | 3.97 | 3.76 | 3.94 | 0.5222 | 0.5934 |
| Disadvantages | 3.47 | 3.56 | 3.45 | 3.58 | 4.10 | 3.51 | 4.2129 | 0.0152 * |
| Assessment | 3.62 | 3.48 | 3.55 | 3.56 | 3.54 | 3.66 | 0.4695 | 0.6254 |

\* $p < 0.05$.

**Table 12.** Standard deviations and Bartlett test statistics when differentiated by academic level.

| Scale | Graduate | | Master | | Ph.D. | | K-Squared | *p*-Value |
|---|---|---|---|---|---|---|---|---|
| | Health | Engineering | Health | Engineering | Health | Engineering | | |
| Technical | 0.83 | 0.83 | 1.01 | 0.76 | 0.81 | 0.82 | 9.1589 | 0.1029 |
| Future | 1.04 | 1.21 | 0.92 | 1.02 | 0.96 | 1.03 | 7.3249 | 0.1976 |
| Disadvantages | 1.24 | 1.10 | 1.38 | 1.22 | 1.07 | 1.38 | 8.9144 | 0.1125 |
| Assessment | 1.20 | 1.18 | 1.29 | 1.30 | 1.21 | 1.34 | 3.0950 | 0.6853 |

### 3.6. Teaching Experience

The descriptive variable of teaching experience has proven to be discriminative in didactic usefulness (within the technical aspects), space shortage (within the disadvantages) and evaluation of the VR training received (within the overall evaluations). Figure 7 shows the frequencies of the answers to these items differentiated by years of teaching experience. In didactic use, the highest proportions of answer 5 are reached by participants with more than 20 years of experience, while answer 4 is more frequent among those with 15 years or less. The answers corresponding to intermediate or low values (1, 2 or 3) also occur

in greater proportion among those with less experience, except for value 1, which only appears among participants with more than 25 years of experience. The highest mean for this item is among participants with between 21 and 25 years of experience (4.75) followed by those with between 6 and 10 years (4.63) and those with more than 25 years (4.62), those with 5 years or less (4.43) and those with between 11 and 15 years of experience (4.35). However, the difference between the highest mean and the lowest mean does not reach two tenths. Furthermore, the ANOVA test does not identify statistically significant differences between them (F = 0.9506, *p*-value = 0.4552).

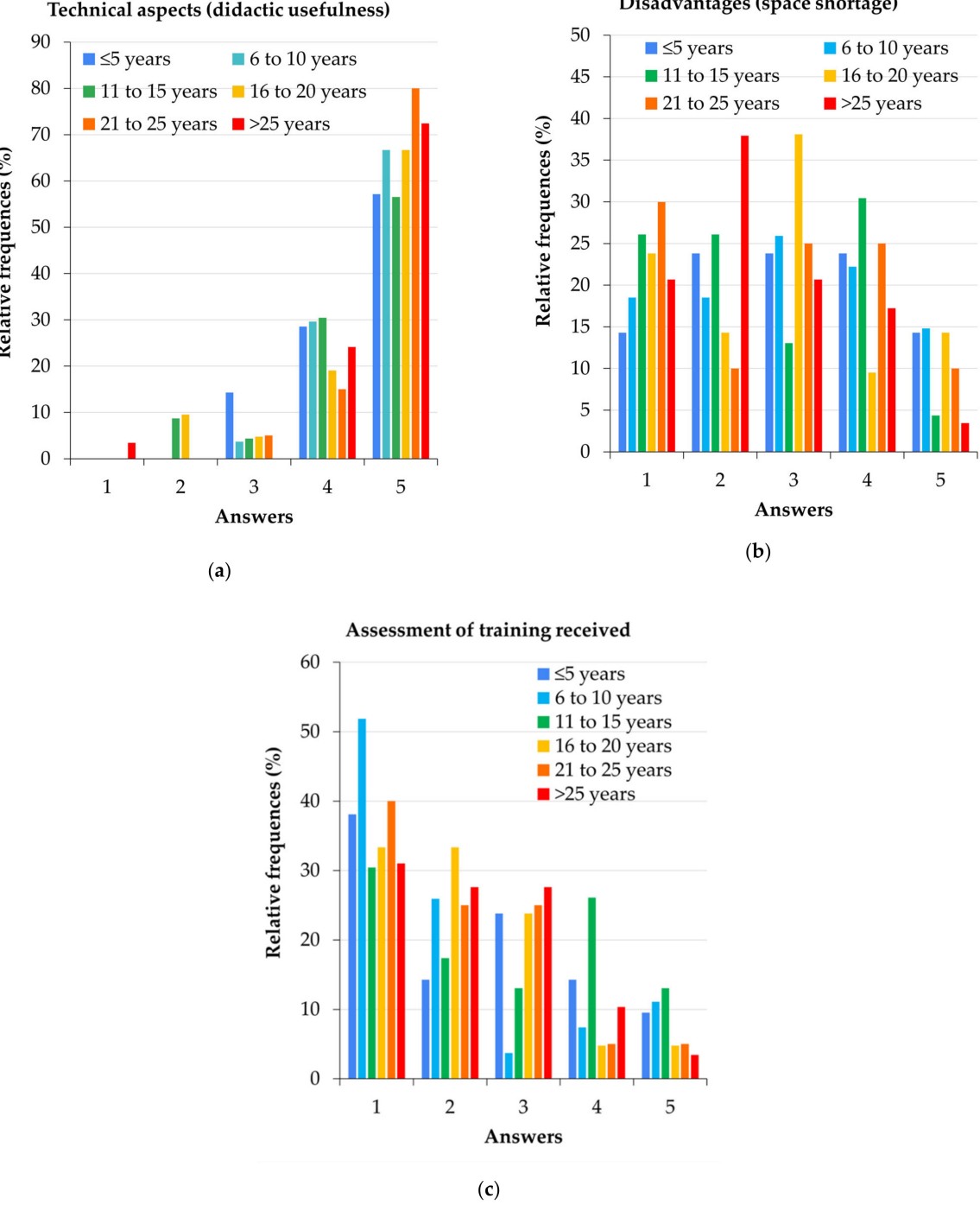

**Figure 7.** Answers on (**a**) didactic usefulness (technical aspects), (**b**) disadvantages (space shortage) and (**c**) assessment of training received by years of teaching experience.

On the inconvenience item, participants with less than 10 years of experience have the highest proportion of answers with value 5 and the lowest proportion of answers with value 1. On average, the participants who most identified the shortage of space as a drawback were those with less experience (3.00 on average) and those with between 6 and 10 years (2.96). Those with more than 25 years of experience perceive this disadvantage less (2.45 on average). The differences between the means are very small (55 hundredths between the lowest and the highest), and the ANOVA test does not identify it as statistically significant (F = 0.7188, *p*-value = 0.6118).

Regarding the evaluation of the training received, in all the ranges of teaching experience, the highest proportions of answers are concentrated in the values 1 to 3, which indicates a general dissatisfaction among the participants with the training they have received on VR. It is noteworthy that participants with more than 15 years of experience concentrate the highest proportion of their answers in values 1 to 3 (decreasing as the value increases). Those with less than 10 years of experience, in fact, concentrate their answers in the value 1. On average, the participants who rate their training worst are those with between 6 and 10 years of experience (2.00), followed by those with between 21 and 25 years (2.10) and those with between 16 and 20 years (2.14). However, again the ANOVA statistics do not allow us to assume these differences between the mean answers as statistically significant (F = 0.8167, *p*-value = 0.5425).

The MANOVA test statistics (Table 13) show that there is a significant gap in the answers on the technical aspects, disadvantages and overall rating scales when the sample is differentiated by years of teaching experience. In technical aspects, participants from Engineering with between 11 and 20 years of experience give a significantly higher mean value than those from Health Sciences. Regarding the identification of drawbacks, the least experienced (less than 15 years of experience) give a higher score in Engineering than in Health Sciences, but this trend is reversed in those with more than 15 years of experience. In terms of the overall rating, the highest score among Engineering teachers is given by those with 11 to 15 years of experience, but the latter are those with the lowest score in Health Sciences.

**Table 13.** Mean answers and MANOVA test statistics when differentiated by teaching experience (where H. means Health Science and E. means Engineering and Architecture).

| Scale | $\leq$5 | | 6 to 10 | | 11 to 15 | | 16 to 20 | | 21 to 25 | | >25 | | MANOVA | *p*-Value |
|---|---|---|---|---|---|---|---|---|---|---|---|---|---|---|
| | H. | E. | H. | E. | H. | E. | H. | E. | H. | E. | H. | E. | | |
| Technical | 4.10 | 4.18 | 3.73 | 4.36 | 3.84 | 4.18 | 4.00 | 4.35 | 4.43 | 4.32 | 4.39 | 4.14 | 5.4536 | 0.0001 * |
| Future | 3.70 | 3.87 | 3.49 | 3.83 | 4.20 | 3.97 | 3.93 | 3.98 | 4.03 | 4.10 | 4.09 | 3.98 | 1.1593 | 0.3277 |
| Disadvantages | 3.33 | 3.51 | 3.47 | 3.83 | 3.00 | 3.49 | 3.69 | 3.32 | 3.93 | 3.63 | 3.89 | 3.49 | 2.8412 | 0.0150 * |
| Assessment | 3.68 | 3.63 | 3.44 | 3.42 | 3.97 | 3.71 | 3.11 | 3.83 | 3.67 | 3.54 | 3.60 | 3.38 | 3.4024 | 0.0047* |

\* *p* < 0.05.

The deviations shown in Table 14 allow us to assume differences in the dispersions of the answers on the scales of technical aspects and future projection. In both scales it is the more experienced teachers who are more homogeneous. However, they do so with almost two tenths of a point difference in technical aspects (with greater dispersion in Health Sciences) and one tenth in future projection (with greater dispersion in Engineering).

The descriptive variable of teaching experience has a certain dependency relationship with the age range variable, in the sense that the growth of one of the variables follows the growth of the other. However, this relationship is moderate, as indicated by Cramer's V parameter (V = 0.7564, *p*-value = 0.0000). For this reason, the results are divergent with respect to the age variable, for which no gaps have been identified in any of the areas of knowledge studied.

**Table 14.** Standard deviations and Bartlett test statistics when differentiated by age range (where H. means Health Science and E. means Engineering and Architecture).

| Scale | ≤5 | | 6 to 10 | | 11 to 15 | | 16 to 20 | | 21 to 25 | | >25 | | K-Squared | *p*-Value |
|---|---|---|---|---|---|---|---|---|---|---|---|---|---|---|
| | H. | E. | H. | E. | H. | E. | H. | E. | H. | E. | H. | E. | | |
| Technical | 0.93 | 0.84 | 0.81 | 0.62 | 1.01 | 0.98 | 1.18 | 0.88 | 0.70 | 0.82 | 0.81 | 0.65 | 28.816 | 0.0024 * |
| Future | 0.92 | 1.04 | 1.06 | 1.05 | 0.69 | 1.14 | 0.86 | 1.13 | 1.16 | 0.97 | 0.88 | 0.98 | 19.828 | 0.0478 * |
| Disadvantages | 1.21 | 1.27 | 1.32 | 1.05 | 1.24 | 1.35 | 1.47 | 1.24 | 1.20 | 1.19 | 1.24 | 1.30 | 9.2944 | 0.5947 |
| Assessment | 1.07 | 1.33 | 1.27 | 1.38 | 1.01 | 1.31 | 1.40 | 1.23 | 1.40 | 1.29 | 1.23 | 1.18 | 15.142 | 0.1761 |

* $p < 0.05$.

## 4. Discussion

At the global level, the highest mean values were obtained for the items of didactic use (within the technical aspects) and those of academic results, motivation and student acceptance (Table 3). All these items receive high or very high mean values with a remarkable homogeneity of answers, as they are the items with the lowest standard deviations. This fact suggests that the participants give importance to the didactic benefits of virtual reality as a resource, above other aspects. The literature supports this conclusion in numerous papers, in which the benefits of using virtual reality are described, both for students' learning outcomes and in certain dimensions of their affectivity linked to their academic performance (such as their motivation or anxiety towards learning) [36–38]. Other studies explain that the didactic success of VR is due to the digital nativeness of the young people of our time [39]. In scientific-technical areas, moreover, VR has been used as a didactic resource more frequently than in other areas [40]. The accumulated experience may also be the reason for the high ratings recorded in this work.

It is also noteworthy that the largest deviations are found in the scales of inconveniences and global ratings. This may be due to the fact that, on the one hand, aspects such as the need for space or technological obsolescence present a great deal of heterogeneity in the answers of the participants (it depends a lot on the use that each one has made of virtual reality), while others, such as costs, are more homogeneous because they are more objective. On the other hand, the dispersion of the overall rating scale may be due to the disparity of aspects covered.

A statistically significant gap has been identified between the perceptions of Health Sciences and Engineering teachers regarding the technical dimension of virtual reality. In addition, it has been found that the knowledge area variable is discriminative for a greater number of items than any other variable, especially in the technical aspects scale. These differences are derived from the specificities of the area of knowledge itself, and no other variable, because the independence between the distributions of the different independent variables has been proven. On the other hand, the area of Health Sciences is no stranger to the use of virtual reality in the classroom because it is not a technical area, but its use is relatively common and widespread [41]. Furthermore, in the area of Engineering there is already a certain tradition in the use of this technique [42]. It can be assumed, therefore, that this gap is due to the greater technological knowledge attributable to engineering teachers, which gives them a more realistic perspective of the technical dimensions of virtual reality. This fact may also be the cause, at least partially, of some results found in the study, such as the fact that Health Sciences teachers older than 65 years present more uniform answers regarding technical aspects of virtual reality.

On the other hand, the fact that the assessment of the disadvantages of VR is the lowest of all the scales in the area of Engineering is in line with some studies that claim that the implementation of virtual reality involves more advantages than disadvantages. Indeed, in [43] it is shown, for example, that VR does not increase costs in technical education in higher education, but rather lowers them, since laboratories can be dispensed with. In [44] it is explained that the development of new software and hardware favors a decrease in the costs of using virtual reality. Thus, the results obtained support the idea that in the

areas of technical knowledge the general idea that the implementation of virtual reality has as one of its main disadvantages the significant increase in costs that it entails does not hold true [45]. Moreover, it allows eliminating some social gaps present among the student body, such as the distance that some students have to travel in order to access laboratories and industries. On the other hand, the didactic effectiveness of the use of VR in Engineering and Architecture (this is one of the technical aspects considered) is also confirmed by some previous works [45]. In the area of Health Sciences, the specialized literature, although abundantly affirming the idea that VR is a technically solvent didactic resource [46–48], focuses much more than in the area of Engineering on highlighting the increase in academic performance that it promotes and the good acceptance that it arouses among students [49,50]. In this sense, the gap identified between the two areas of knowledge regarding the valuation of the technical aspects of RLV is in line with confirming the theses of these previous studies.

As for the preferences between IVR, NIVR, AR and MR, it is again the area of knowledge that is the most discriminating variable. The results reveal that teachers in the area of Engineering value the immersive aspects of virtual reality more than those in Health Sciences. This fact can be verified by the high frequency of the highest score given by Engineering teachers to the immersive technical aspect and to the future projection of MR, which combines the benefits of the immersive and non-immersive dimensions of VR. This is probably due to the fact that IVR and RM involve additional elements to the computer (such as glasses or gloves), which increases their technical complexity, and are more naturally adapted to the contents of Engineering and Architecture courses. In fact, some studies show that FTI allows better learning results than NIVR when applied to the teaching of technical concepts, such as kinematics [51]. In the area of Health Sciences, although both immersive and non-immersive technologies are used for didactic purposes depending on the specific object of study, the specialized literature explains that NIVR and AR are more widespread and that, in some specific areas, such as cognitive rehabilitation, among others, it is necessary to intensify the analysis of the applicability of FTI and its benefits [52]. It is therefore detected that the aforementioned technical skill gap between the two areas considered also influences the preference for the immersive dimension of VR.

The fact that no gender gap was identified in either of the two areas of knowledge analyzed is in contradiction with some studies that have found gender gaps in the development of digital competencies in Latin America [53]. This discrepancy may be due to the fact that this study focuses on the assessment specifically of virtual reality and explores only university teachers of Engineering and Health Sciences. In this specific area, the gap is diluted and the perspective expressed by the teachers is more similar to that which some studies attribute to the North American area on the use of ICT as a teaching resource [54].

In this study, no gaps by age were identified in the assessment of RV. This observation is novel with respect to the results of other specialized studies, because, as far as we have been able to explore, there are no publications that analyze differences by age or experience in this regard, although there are works that look for such differences in terms of digital competencies or use of ICT in the higher education classroom. These latter studies do not identify gaps in the use and valuation of the different digital didactic resources considered derived from the age of the participants [55,56]. In this sense, the results of the present article are in line with other previous work.

On the other hand, previous studies had equated the variables age and experience in the identification of gaps in the use and valuation of virtual reality [56]. The results obtained in this study prove that the two variables should be distinguished, because, unlike age, teaching experience provides important differences when crossed with the area of knowledge. In particular, the most experienced Engineering teachers are those who best value virtual reality as a teaching resource in several of its dimensions and who identify fewer drawbacks. This trend also occurs in Health Sciences, although with somewhat less intensity and except that the most experienced teachers here give a high score to the disadvantages. This may again be due to the differences in technological knowledge and

to a certain resistance that the most experienced offer to the incorporation of new didactic resources, especially when this incorporation requires intense training on their part.

As for the academic level, it has been shown that Engineering teachers who have reached a PhD degree are those who best value the technical aspects of virtual reality and give less importance to its drawbacks. Assuming that this population is distinguished by being, of all those considered, the one that brings together the greatest technological knowledge, this fact can be placed in the cause of the identified gap. In addition, Health Sciences teachers with lower academic levels (presumably, more poorly trained in technological knowledge) are those who give lower scores to the technical aspects of virtual reality.

Possible future lines of research include: (i) the identification and description of possible discriminatory reasons; and (ii) the extension to other areas of knowledge and the identification and description of the differences that exist between these areas with regards to teachers' evaluations of virtual reality. Related to the latter, it would be interesting to explore the evaluations of university students and to discuss this with the perception of the teaching staff.

## 5. Conclusions

This paper has explored the valuation that Latin American university teachers make of virtual reality as a teaching resource. The sample of teachers has been focused on the areas of Health Sciences and Engineering and Architecture, as these are the fields that can potentially make the most intense use of this resource in the classroom. Thus, the area of knowledge is the main variable of the study, and the rest have been analyzed by cross-referencing them with it.

A significant gap has been identified by area of knowledge in the valuation of the technical aspects of virtual reality. In addition, the area of knowledge is a discriminant variable for a variety of items studied. Specifically, teachers of Health Sciences give a worse evaluation of this technical dimension than teachers of Engineering, probably due to the fact that their training gives them a weaker knowledge of technological aspects. It can be concluded that technological knowledge has a positive influence on the identification and valuation of the technical dimensions of virtual reality, from which it follows that it is advisable to train non-specialist teachers in this type of dimension.

Unlike what happens in general, when focusing on Health Sciences and Engineering, the gender and age gaps are diluted when these variables are crossed with the area of knowledge. However, notable and statistically significant differences are found when differentiating by academic level and length of teaching experience. The differences by academic level are again centered on technical aspects and also on the identification of drawbacks, and suggest that knowledge in the technological area of those who suffer most from it would increase this assessment. The gap in terms of years of teaching experience affects Health Sciences teachers more (again, those least trained in technology) and may be due to a certain effect of opposition to a change in teaching methodology. It is worth suggesting the implementation by universities of specific training plans in digital learning resources for the most experienced teaching staff.

**Author Contributions:** Conceptualization, D.V.; methodology, D.V. and Á.A.-S.; validation, D.V., Á.A.-S. and P.F.-A.; formal analysis, D.V., Á.A.-S., J.E. and P.F.-A.; data curation, D.V. and Á.A.-S.; writing—original draft preparation, D.V., Á.A.-S., J.E. and P.F.-A.; writing—review and editing, D.V., Á.A.-S., J.E. and P.F.-A. All authors have read and agreed to the published version of the manuscript.

**Funding:** This research received no external funding.

**Institutional Review Board Statement:** All participants were informed about the anonymous nature of their participation, why the research is being conducted, how their data will be used and that under no circumstances would their data be used to identify them. The protocol was approved by the Ethics Committee of the Project "Influence of COVID-19 on teaching: digitization of laboratory practices at UCAV" (17 May 2021).

**Informed Consent Statement:** Not applicable.

**Data Availability Statement:** The data are not publicly available because they are part of a larger project involving more researchers. If you have any questions, please ask the contact author.

**Conflicts of Interest:** The authors declare no conflict of interest.

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
