# Peer review of "Assessment of Virtual Reality as a Didactic Resource in Higher Education"

_sustainability, doi:10.3390/su132212730_

Round 1
Reviewer 1 Report
The work is of great interest to the scientific and educational field. It is well written, with an easy understanding. Adequate length and the data are well presented.
As the only element of improvement, I would propose to indicate how many participants were contacted initially, how many did not answer or refused to participate, and the reasons (if known).
Otherwise, it's a great job.
Author Response
Please, find the enclosed document for the answer.

Reviewer 2 Report
The manuscript is well structured and developed according to the methodological approach.
It is well supported theoretically, with recent and relevant references about the phenomenon under study.
The methodology is consistent with the objectives pursued and the representation of the results is clear.
Finally, the discussion is carried out in contrast with previous literature, allowing us to see the scope of the study and its contribution to the field of knowledge.
Author Response
Thank you very much for your kind comments
Reviewer 3 Report
A very well written article, methodologically clear and sound statistical data analysis.
Author Response

(The authors gave the same response as above.)

Reviewer 4 Report
This work is pretty good!
In the section of Introduction, the authors need to introduce the practical value of the findings in higher education.
The figures in this manuscript can help readers to understand what the authors do, and the research design will provide some new ideas for future research.
Author Response

(The authors gave the same response as above.)
